# Non-communicable diseases and resistant tuberculosis, a growing burden among people living with HIV in Eastern Kenya

Patrick Kiogora Muriuki[1]*, Musa Otieno Ngayo[2], Moses Njire[3], Juster Mungiria[2], Winfred Asiko Nyanya[2], Daniel Owuor[2], Perpetual Ndung'u[1]

1 Department of Medical Laboratory Science, Jomo Kenyatta University of Agriculture and Technology, Kenya, 2 Centre of Microbiology Research, Kenya Medical Research Institute, Nairobi, Kenya, 3 Department of Biochemistry and Molecular Biology, Jomo Kenyatta University of Agriculture and Technology, Kenya

* patrick.kiogora@gmail.com

## Abstract

Human Immunodeficiency Virus (HIV) and tuberculosis (TB) continue to pose a significant health burden in Kenya. Countries with the highest rates of people living with HIV (PLWH) also have a high prevalence of non-communicable diseases (NCDs), including type 2 diabetes (T2D) and hypertension (HPT). This study evaluated the burden and factors associated with T2D, HPT, and TB, including resistant strains among PLWH receiving antiretroviral therapy (ART) in Eastern Kenya. Blood and sputum samples, and baseline information were collected from 280 consenting PLWH. The participants' blood pressure (BP), glycated hemoglobin (HbA1c), CD4 cell counts, HIV viral load, full blood count, blood chemistry, and Rifampicin resistance were assessed. The mean (SD) age of the participants was 35.6 (±9.8) years, and a median (IQR) duration of living with HIV of 7 (4 –8) years. Most participants, 179 (63.9%), were HIV mono-infected, with 58 (20.7%) HIV/TB, 42 (15%) HIV/T2D, and 33 (11.8%) HIV/HPT dual comorbidities reported. Triple comorbidities reported included 18 (6.4%) HIV/T2D/HPT, 9 (3.2%) HIV/TB/T2D, and 9 (3.2%) HIV/TB/HPT, with 4 (1.4%) HIV/TB/T2D/HPT quadruple comorbidity reported. Six (2.1%) multidrug-resistant TB coinfections were detected. In multivariate analyses, being on ARV only (aOR 0.5; 95% CI 0.4 – 0.6, p = 0.0001) and achieving virological suppression (aOR 0.8; 95% CI 0.6 – 0.9, p = 0.017) were protective against HIV/TB coinfection. Previous hospital admission (aOR 1.2; 95% CI 1.1 – 1.4, p = 0.049) and previous TB infection (aOR 1.6; 95% CI 1.0 – 3.0, p = 0.034) were associated with HIV/TB coinfection.

The PLWH in Eastern Kenya continues to experience a syndemic of NCDs and TB, including resistant strains. Consistent adherence to ART is crucial for achieving viral suppression; these are protective against NCDs and TB among PLWH. The findings highlight the necessity of integrating NCD management with HIV and TB treatment programs in Kenya.

**Data availability statement:** The data has been uploaded as Supporting information.

**Funding:** The study was supported by student Loan.

**Competing interests:** The authors have declared that no competing interests exist.

## Introduction

Human Immunodeficiency Virus (HIV) is still a public health threat in Kenya, and by 2023, about 1.4 million people were living with HIV (PLWH) [1]. Since the commencement of antiretroviral therapy (ART) in the 1990s and the subsequent scale-up, HIV in Kenya is currently a manageable chronic condition. About 94% of PLWH are presently on ART, with 94% achieving virological suppression (< 200 copies/mL). Increased life expectancy among PLWH has become associated with the concurrent increase in the prevalence of both infectious diseases, such as tuberculosis (TB) as well as non-communicable diseases (NCDs) such as obesity, hypertension (HPT), and type 2 diabetes (T2D) [2]. There is a co-existence of both chronic infectious and non-infectious diseases, both exacerbated by a greatly unequal society, socioeconomic status, and other social determinants [3]. TB is still a major cause of ill health and death globally. Before the coronavirus (COVID-19) pandemic, TB was ranked as the number one leading cause of fatalities, ranking above HIV/AIDS [4]. In Africa, by 2023, Kenya was listed among the countries with the highest burden for TB, HIV-associated TB, and multidrug-resistant or rifampicin-resistant TB (MDR/RR-TB) [4]. The interaction between TB and HIV infection is a key driver for the increased TB burden in Kenya, with a prevalence of 237 per 100,000 population and new estimated cases of 288 per 100,000 population [4]. In 2022, over 7,000 and 8900 deaths among HIV-negative and HIV-positive individuals, respectively, were attributed to TB infection [4].

For a while, in most countries in Sub-Saharan Africa, NCDs have been regarded as diseases of the rich, and T2D and HPT are becoming increasingly prevalent [5,6]. In Kenya, NCDs are estimated to account for 39% of all deaths, with HPT and T2D accounting for 13.8% and 3.5% of these deaths, respectively [7,8]. With enhanced socioeconomic status, minimized morbidity and mortality owing to infectious diseases, rapid development, and changes in the way of living and dietary habits, developing countries are increasingly reporting surges in NCDs [9]. The TB infection only becomes a disease when a person's immune system becomes compromised. The NCDs are shown to compromise the immunity of affected persons. Extended dependence on medication, diabetes, smoking, and consumption of alcohol are the major risk factors for TB [10]. Chronic TB infections have been shown to contribute to the development of HPT, and a reverse association may exist, such that HPT may lead to an increased risk of developing TB [11]

Comorbidity of TB with NCDs and other communicable diseases, such as HIV/AIDS, is prevalent globally in regions that are highly endemic for TB; therefore, integrated and effective responses are essential to tackle these comorbidities [4]. Inevitably, tackling TB also requires tackling NCDs as a matter of urgency [12]. Even though the rate of active TB is much higher in specific groups, such as those on immunosuppressants or with HIV infection, as well as patients with comorbidities like T2D or HPT [13]. The risk of progression from latent infection to active disease in the general population is influenced by factors such as tobacco smoking, alcohol consumption, physical inactivity, and malnutrition [10]. Studies must continue to provide information on the comorbidity of TB and NCDs, especially in developing

countries, to strategize on the mechanisms to reduce their burden, more so among PLWH. Inevitably, understanding the associations of combined NCD risk factors with TB risk could offer information enabling the translation of epidemiological results into prevention strategies. This study determined the burden and predictors of NCDs and TB among PLWH in Eastern Kenya.

## Methods

### Study design

This cross-sectional study was undertaken among PLWH from 10th June 2021–9th June 2022 in Nyambene Sub-County Hospital in Eastern Kenya.

### Eligibility criteria

Participants enrolled in this study were those consenting, aged above 18 years, provided sputum and blood samples, underwent a clinical medical examination, and received ART treatment and care at Nyambene Sub-County Hospital in Meru County in Eastern Kenya.

### Informed consent

Written informed consent was obtained from all participants before enrolment into the study.

### Sample size

Sample size calculation used the formula described by Lemashow [14] based on population proportion estimation. Setting the alpha (α) at 0.05, critical value (z) based on the desired confidence level (∝ at 1.96), and the prevalence (p) of TB and NCDs among PLWH prevalence of 19.8% [15], a total of 280 eligible PLWH were enrolled with 15% added to cover for lost to follow up.

### Ethical statement

This research was carried out according to the basic principles in the Guidance for Good Clinical Practice and those in the Declaration of Helsinki (Edinburg, October 2000). This protocol and all the study tools, including informed consent forms, were reviewed, and permission was granted by the Jomo Kenyatta University of Agriculture and Technology (JKUAT) Ethical Review Committee (ERC) (JKU/IERC/02316/0134). The National Commission for Science, Technology & Innovation (NACOSTI) issued the study license (License No. NACOSTI/21/12006). Written informed consent was obtained from all participants before enrollment.

### Information collection

The participants' baseline characteristics were collected using structured, face-to-face interviews.

Five ml of blood samples were drawn from each participant into an ethylenediaminetetraacetic acid (EDTA) tube and serum separating tube (SST). The blood samples collected in the EDTA tubes were used for hematological and immunological analysis, and the blood collected in the SSTs was used for clinical chemistry analysis. If not used immediately, these samples were stored at −80°C.

Two early morning sputum samples were collected from each participant. An ultrasonic nebulizer method was used to stimulate sputum production for those unable to provide. The samples were stored at 4°C before laboratory analysis.

Each participant's blood pressure (BP) was measured using a Citizen Digital Arm Blood Pressure Monitor CH-503 (Citizen Systems Japan CO., LTD). Blood pressure (BP) readings with systolic BP of ≥140 mmHg and/or diastolic BP of ≥90 mmHg were classified as hypertensive.

### Laboratory analysis

**Glycated hemoglobin (HbA1C) testing.** Glycated hemoglobin (HbA1C) testing was used to determine the average blood sugar concentration and the extent of carbohydrate imbalance over the previous two to three months. The HbA1C was measured using the SD A1cCare analyzer (SD Biosensor, Inc.). The test results were interpreted as follows: < 8% HbA1c (64 mmol/mol) indicated less stringent glycemic control, < 7% HbA1c (53 mmol/mol) signified general glycemic control, while <6.5% HbA1c (48 mmol/mol) indicated more stringent glycemic control. Participants with HbA1c levels ranging between 5.7% and 6.4% (39–46 mmol/mol) were at an increased risk for diabetes.

### Detection of tuberculosis

**Microscopy.** The Ziehl-Neelsen (ZN) staining method using the first sputum samples was used to detect acid-fast bacillus, which stained red. Liquid sputum samples were first concentrated by centrifugation at 3000 rpm for 15 minutes and ZN staining the precipitate. Thick or purulent sputum samples were diluted using N acetyl-L-cysteine ZN staining. ZN-stained smears were graded according to the International Union against Tuberculosis and Lung Disease scale. TB-positive smears were those graded as scanty or higher scores.

**Detection of MTB/RIF.** GeneXpert MTB/RIF diagnostic system (Cepheid, Sunnyvale, CA, USA) using the second sputum sample was used to detect either TB-positive rifampicin-resistant strains or TB-positive non-resistant strains as described by Fouda et al., [16].

### Immunological and Biochemical assessment

The CD4 count was measured using a BD FACSpresto (Becton Dickinson, BD Biosciences, San Jose, USA). A complete hemogram was measured using a Medonic M - series M32 hematology analyzer (Boule Clinical Diagnostic Solutions, Inc.). The Aspartate transferase (AST), Alanine transaminase (ALT), and Creatinine were measured using the DRI-CHEM NX500 dry chemistry analyzer (Fujifilm, Czech Republic). These tests were done using the procedures provided by the manufacturers.

### Determination of plasma HIV-1 RNA concentration

The HIV-1 RNA concentration (viral load) was measured using the Generic HIV Viral Load assay (Biocentric, Bandol-France). Briefly, viral RNA was extracted from 1 mL plasma samples using a QIAmp viral RNA mini kit (Qiagen Inc., USA). Using the Biocentric assay, 10 µL of RNA samples were amplified using the ABI Prism 7300 Sequence Detection System (Applied Biosystems). The amplification conditions were as follows: initial heating at 50°C for 10 minutes and 95°C for 5 minutes, followed by 50 cycles of 95°C for 15 seconds and 60°C for 1 minute. The test had a detection limit of 40 HIV-1 RNA copies/mL.

### Statistical analysis

Baseline variables were summarized with descriptive statistics. Bivariate logistic regression analyses were performed to assess the association of each variable with the syndemic of NCDs and TB among PLWH. The multivariate logistic regression model was then utilized to examine the independent associations between the identified correlates and various comorbidities of NCDs and TB, including resistant strains among PLWH. Results (S1 Data) were analyzed using R software version 4.1.2 at a significance level of $p \leq 0.05$.

## Results

### Participants' baseline characteristics (n=280)

Table 1 summarizes the participants' baseline characteristics. Of the 280 PLWH recruited from Nyambene region, Meru County, their mean (SD) age was 35.6 (±9.8) years, 146 (52.1%) were male, 91 (32.5%) had no formal education

**Table 1. Baseline characteristics of the study population (n = 280 patients).**

| Variables | All Patients | COMORBIDITY TYPE | | | | | | | |
|---|---|---|---|---|---|---|---|---|---|
| | | HIV/TB | HIV/MDR-TB | HIV/T2D | HIV/HPT | HIV/T2D/HPT | HIV/TB/T2D | HIV/TB/HPT | HIV/TB/T2D/HPT |
| | n (280) | n (%) 58(20.7) | n (%) 6(2.1) | n (%) 42(15) | n (%) 33(11.8) | n (%) 18(6.4) | n (%) 9(3.2) | n (%) 9(3.2) | n (%) 4 (1.4) |
| **Age** | | | | | | | | | |
| **Mean (± SD)** | 35.6 (±9.8) | | | | | | | | |
| 18 -30 | 82 | 12 (14.6) | 0 | 1 (1.2) | 1 (1.2) | 0 | 1 (1.2) | 0 | 0 |
| 31 - 40 | 121 | 27 (22.3) | 6 (5) | 10 (8.3) | 4 (3.3) | 0 | 2 (1.7) | 1 (0.8) | 0 |
| 41 - 50 | 53 | 11 (20.8) | 0 | 13 (24.5) | 7 (13.2) | 2 (4) | 2 (3.8) | 2 (3.8) | 0 |
| ≥51 | 24 | 7 (29.2) | 0 | 18 (75) | 21 (87.5) | 16 (67) | 4 (16.7) | 6 (25) | 4 (16.7) |
| *P value* | | *0.417* | ***0.047*** | ***0.001*** | ***0.001*** | ***0.001*** | ***0.001*** | ***0.001*** | ***0.001*** |
| **Gender** | | | | | | | | | |
| Male | 146 | 29 (19.9) | 3 (2.1) | 25 (17.1) | 27 (18.5) | 16 (11) | 5 (3.4) | 7 (4.8) | 4 (2.7) |
| Female | 134 | 29 (21.6) | 3 (2.2) | 17 (12.7) | 6 (4.5) | 2 (2) | 4 (3) | 2 (1.5) | 0 |
| *P value* | | *0.714* | *0.915* | *0.299* | ***0.001*** | ***0.001*** | *0.835* | *0.118* | *0.054* |
| **Education level** | | | | | | | | | |
| Primary | 47 | 8 (17) | 1 (2.1) | 7 (14.9) | 7 (14.9) | 4 (9) | 0 | 0 | 0 |
| Secondary | 88 | 23 (26.1) | 1 (1.1) | 15 (17) | 11 (12.5) | 4 (5) | 4 (4.5) | 6 (6.8) | 1 (1.1) |
| Tertiary | 54 | 12 (22.2) | 1 (1.9) | 7 (13) | 4 (7.4) | 2 (4) | 2 (3.7) | 1 (1.9) | 1 (1.9) |
| Non-Formal | 91 | 15 (16.5) | 3 (3.3) | 13 (14.3) | 11 (12.5) | 8 (9) | 3 (3.3) | 2 (2.2) | 2 (2.2) |
| *P value* | | *0.387* | *0.795* | *0.919* | *0.686* | *0.500* | *0.551* | *0.119* | *0.756* |
| **Occupation** | | | | | | | | | |
| Business | 106 | 18 (17) | 2 (1.9) | 18 (17) | 17 (16) | 10 (9) | 2 (1.9) | 4 (3.8) | 1 (0.9) |
| Employed | 57 | 17 (29.8) | 1 (1.8) | 8 (14) | 5 (8.8) | 2 (4) | 3 (5.3) | 2 (3.5) | 1 (1.8) |
| Farming | 28 | 6 (21.4) | 0 | 1 (3.6) | 3 (10.7) | 1 (4) | 0 | 0 | 0 |
| Unemployed | 89 | 17 (19.1) | 3 (3.4) | 15 (16.9) | 8 (9) | 5 (6) | 4 (4.5) | 3 (3.4) | 2 (2.2) |
| *P value* | | *0.269* | *0.723* | *0.324* | *0.385* | *0.411* | *0.429* | *0.787* | *0.789* |
| **Duration post ART initiation (Years)** | | | | | | | | | |
| **Median (IQR)** | 7 (4 − 8) | | | | | | | | |
| 1–5 | 94 | 22 (23.4) | 2 (2.1) | 15 (16) | 12 (12.8) | 5 (5) | 2 (2.1) | 5 (5.3) | 1 (1.1) |
| 6–10 | 165 | 30 (18.2) | 4 (2.4) | 28 (17) | 19 (11.5) | 11 (7) | 7 (4.2) | 4 (2.4) | 3 (1.8) |
| > 11 | 21 | 6 (28.6) | 0 | 2 (9.5) | 2 (9.5) | 2 (10) | 0 | 0 | 0 |
| *P value* | | *0.397* | *0.77* | *0.506* | *0.904* | *0.762* | *0.446* | *0.306* | *0.752* |
| **Changed ARV** | | | | | | | | | |
| Yes | 196 | 46 (23.5) | 5 (2.6) | 30 (15.3) | 25 (12.8) | 13 (7) | 7 (3.6) | 8 (4.1) | 3 (1.5) |
| No | 84 | 12 (14.3) | 1 (1.2) | 12 (14.3) | 8 (9.5) | 5 (6) | 2 (2.4) | 1 (1.2) | 1 (1.2) |
| *P value* | | *0.082* | *0.77* | *0.827* | *0.442* | *0.832* | *0.605* | *0.209* | *0.826* |
| **Hospital Admission** | | | | | | | | | |
| Yes | 28 | 12 (42.9) | 6 (21.4) | 7 (25) | 26 (92.9) | 5 (18) | 4 (14.3) | 5 (17.9) | 4 (14.3) |
| No | 252 | 46 (18.3) | 0 | 35 (13.9) | 7 (2.8) | 13 (5) | 5 (2) | 4 (1.6) | 0 |
| *P value* | | ***0.002*** | ***0.001*** | *0.118* | ***0.022*** | ***0.009*** | ***0.0001*** | ***0.0001*** | ***0.0001*** |
| **Missed taking current ART** | | | | | | | | | |
| Yes | 104 | 23 (22.1) | 3 (2.9) | 3 (2.9) | 6 (5.8) | 1 (1) | 1 (1) | 2 (1.9) | 0 |
| No | 176 | 35 (19.9) | 3 (1.7) | 39 (22.2) | 27 (15.3) | 17 (10) | 8 (4.5) | 7 (4) | 4 (2.3) |
| *P value* | | *0.657* | *0.159* | ***0.001*** | ***0.016*** | ***0.004*** | *0.1* | *0.346* | *0.121* |

*(Continued)*

Table 1. (Continued)

| Variables | All Patients | COMORBIDITY TYPE | | | | | | | |
|---|---|---|---|---|---|---|---|---|---|
| | | HIV/TB | HIV/MDR-TB | HIV/T2D | HIV/HPT | HIV/T2D/HPT | HIV/TB/T2D | HIV/TB/HPT | HIV/TB/T2D/HPT |
| | n (280) | n (%) 58(20.7) | n (%) 6(2.1) | n (%) 42(15) | n (%) 33(11.8) | n (%) 18(6.4) | n (%) 9(3.2) | n (%) 9(3.2) | n (%) 4 (1.4) |
| **No of times Missed taking current ART** | | | | | | | | | |
| Once | 76 | 18 (23.7) | 3 (3.9) | 2 (2.6) | 2 (2.6) | 0 | 1 (1.3) | 1 (1.3) | 0 |
| Twice | 13 | 3 (23.1) | 0 | 0 | 1 (7.7) | 1 (8) | 0 | 1 (7.7) | 0 |
| Thrice or more | 15 | 2 (13.3) | 0 | 1 (7.7) | 3 (20) | 0 | 0 | 0 | 0 |
| None | 176 | 35 (19.9) | 3 (1.7) | 39 (22.2) | 27 (15.3) | 17 (10) | 8 (4.5) | 7 (4) | 4 (2.3) |
| *P value* | | *0.796* | *0.582* | ***0.001*** | ***0.024*** | ***0.025*** | *0.421* | *0.529* | *0.494* |
| **Reasons for missing taking current ART** | | | | | | | | | |
| Experiencing Side Effects | 20 | 5 (25) | 3 (15) | 0 | 3 (15) | 0 | 0 | 1 (5) | 0 |
| Forgot to take drugs at stipulated time | 72 | 13 (18.1) | 0 | 3 (4.2) | 3 (4.2) | 1 (1) | 1 (1.4) | 1 (1.4) | 0 |
| Lapse of drugs supply | 12 | 5 (41.7) | 0 | 0 | 0 | 0 | 0 | 0 | 0 |
| Not applicable | 176 | 35 (19.9) | 3 (1.7) | 39 (22.2) | 27 (15.3) | 17 (10) | 8 (4.5) | 7 (4) | 4 (2.3) |
| *P value* | | *0.282* | ***0.001*** | ***0.001*** | ***0.047*** | ***0.040*** | *0.418* | *0.636* | *0.494* |
| **Types of adverse effects** | | | | | | | | | |
| Bloating pain or gas in stomach | 14 | 2 (14.3) | 0 | 1 (7.1) | 1 (7.1) | 0 | 0 | 0 | 0 |
| Diarrhea | 66 | 15 (22.7) | 3 (4.5) | 3 (4.5) | 5 (7.6) | 1 (2) | 1 (1.5) | 1 (1.5) | 0 |
| Fatigue | 13 | 5 (38.5) | 0 | 3 (23.1) | 0 | 0 | 2 (15.4) | 0 | 0 |
| Headache | 11 | 6 (54.5) | 3 (27.3) | 0 | 0 | 0 | 0 | 0 | 0 |
| Nausea or Vomiting | 14 | 0 | 0 | 1 (7.1) | 1 (7.1) | 0 | 0 | 0 | 0 |
| Skin Problems/ Fungal Infections | 53 | 11 (20.8) | 0 | 14 (26.4) | 9 (17) | 6 (6) | 1 (1.9) | 1 (1.9) | 0 |
| Weight Gain | 18 | 3 (16.7) | 0 | 3 (16.7) | 3 (16.7) | 0 | 0 | 1 (5.6) | 0 |
| Weight Loss or wasting | 83 | 16 (19.3) | 0 | 17 (20.5) | 14 (16.9) | 11 (13) | 5 (6) | 6 (7.2) | 4 (4.8) |
| *P value* | | ***0.04*** | ***0.001*** | ***0.022*** | *0.302* | ***0.029*** | *0.13* | *0.407* | *0.221* |
| **Duration of cough experienced** | | | | | | | | | |
| >3 Weeks | 143 | 33 (23.1) | 0 | 31 (21.7) | 12 (8.4) | 6 (4) | 7 (4.9) | 5 (3.5) | 2 (1.4) |
| 4-10 Weeks | 124 | 24 (19.4) | 6 (4.8) | 11 (8.9) | 21 (17) | 0 | 2 (1.6) | 4 (3.2) | 2 (1.6) |
| >11 Weeks | 13 | 1 (7.7) | 0 | 0 | 0 | 12 (92) | 0 | 0 | 0 |
| *P value* | | *0.374* | ***0.021*** | ***0.004*** | *0.181* | *0.312* | *0.253* | *0.791* | *0.896* |
| **Previous tuberculosis diagnosis** | | | | | | | | | |
| Yes | 274 | 52 (19) | 6 (2.2) | 42 (15.3) | 33 (12) | 18 (7) | 9 (3) | 9 (3) | 0 |
| No | 6 | 6 (100) | 0 | 0 | 0 | 0 | 0 | 0 | 4 (66.7) |
| *P value* | | ***0.001*** | ***0.001*** | *0.298* | *0.365* | *0.516* | *0.652* | *0.652* | *0.766* |
| **Previous diabetes diagnosis** | | | | | | | | | |
| Yes | 258 | 54 (20.9) | 0 | 22 (8.5) | 13 (5) | 5 (2) | 4 (1.6) | 4 (1.6) | 4 (1.6) |
| No | 22 | 4 (18.2) | 6 (27.3) | 20 (90.9) | 20 (90.9) | 13 (59) | 5 (22.7) | 5 (22.7) | 0 |
| *P value* | | *0.76* | *0.47* | ***0.001*** | ***0.001*** | ***0.001*** | ***0.001*** | ***0.001*** | ***0.001*** |
| **Previous hypertension diagnosis** | | | | | | | | | |
| Yes | 22 | 4 (18.2) | 0 | 9 (40.9) | 18 (81.8) | 9 (41) | 0 | 4 (18.2) | 0 |
| No | 258 | 54 (20.9) | 6 (2.3) | 33 (12.8) | 15 (5.8) | 9 (3) | 9 (3.5) | 5 (1.9) | 4 (1.6) |

*(Continued)*

**Table 1.** (Continued)

| Variables | All Patients | COMORBIDITY TYPE | | | | | | | |
|---|---|---|---|---|---|---|---|---|---|
| | | HIV/TB | HIV/MDR-TB | HIV/T2D | HIV/HPT | HIV/T2D/HPT | HIV/TB/T2D | HIV/TB/HPT | HIV/TB/T2D/HPT |
| | n (280) | n (%) 58(20.7) | n (%) 6(2.1) | n (%) 42(15) | n (%) 33(11.8) | n (%) 18(6.4) | n (%) 9(3.2) | n (%) 9(3.2) | n (%) 4 (1.4) |
| *P value* | | *0.76* | *0.47* | ***0.001*** | ***0.001*** | ***0.001*** | *0.373* | ***0.001*** | *0.556* |
| **Family history of hypertension** | | | | | | | | | |
| Yes | 114 | 32 (28.1) | 0 | 38 (33.3) | 29 (25.4) | 15 (13) | 5 (4.4) | 8 (7) | 3 (2.6) |
| No | 166 | 26 (15.7) | 6 (3.6) | 4 (2.4) | 4 (2.4) | 3 (2) | 4 (2.4) | 1 (0.6) | 1 (0.6) |
| *P value* | | *0.314* | ***0.015*** | *0.184* | ***0.001*** | ***0.001*** | *0.357* | ***0.003*** | *0.166* |
| **Family history of diabetes** | | | | | | | | | |
| Yes | 138 | 34 (24.6) | 6 (4.3) | 21 (15.2) | 20 (14.5) | 18 (13) | 9 (6.5) | 5 (3.6) | 4 (2.9) |
| No | 142 | 24 (16.9) | 0 | 21 (14.8) | 13 (9.2) | 0 | 0 | 4 (2.8) | 0 |
| *P value* | | ***0.002*** | ***0.003*** | ***0.001*** | *0.166* | ***0.001*** | ***0.002*** | *0.702* | ***0.041*** |
| **Alcohol Uptake** | | | | | | | | | |
| Yes | 97 | 16 (16.5) | 3 (3.1) | 17 (17.5) | 12 (12.4) | 10 (10) | 2 (2.1) | 2 (2.1) | 1 (1) |
| No | 183 | 42 (23) | 3 (1.6) | 25 (13.7) | 21 (11.5) | 8 (4) | 7 (3.8) | 7 (3.8) | 3 (1.6) |
| *P value* | | *0.205* | *0.424* | *0.386* | *0.825* | *0.054* | *0.426* | *0.426* | *0.683* |
| **Tobacco use** | | | | | | | | | |
| Yes | 78 | 15 (19.2) | 3 (3.8) | 6 (7.7) | 10 (12.8) | 3 (4) | 2 (2.6) | 4 (5.1) | 1 (1.3) |
| No | 202 | 43 (21.3) | 3 (1.5) | 36 (17.8) | 23 (11.4) | 15 (7) | 7 (3.5) | 5 (2.5) | 3 (1.5) |
| *P value* | | *0.703* | *0.221* | ***0.033*** | *0.739* | *0.274* | *0.701* | *0.259* | *0.898* |

T2D- Type 2 Diabetes, TB – Tuberculosis, HTP- Hypertension, MDR – Multidrug-resistant, n - number; % - percentage; IQR - Interquartile range; SD Standard Deviation; *P value* - Chi-Square Test.

level, while 207(73.9%) were married. The majority of the participants, 169 (82.4%) were on the first-line ART regimen, 179(63.9%) were on a combination therapy of dolutegravir/lamivudine/tenofovir (DTG/3TC/TDF), while 4(1.4%) were on four different drug combinations (ARV + Anti-TB + antidiabetic + antihypertensive). The median (IQR) duration living with HIV was 7 (4 –8) years, and the median (IQR) duration on ART was 6 (4 –7). About 28 (10%) of the participants were hospitalized based on the current conditions, with a 104 (37.1%) non-adherence rate. About half, 143 (51.1%) of the participants had cough lasting > 3 weeks, with 6 (2.1%), 22 (7.9%), and 22 (7.9%) previously diagnosed with tuberculosis, diabetes, and hypertension, respectively. There were 114 (50.7%) and 138 (50.7%) participants with a family history of hypertension and diabetes, respectively. Of the 280 recruited participants, 97 (34.6%) were taking alcohol, while 78 (27.9%) were either smoking or taking tobacco products.

## Immuno-pathological laboratory outcomes

Table 2 summarizes the immune-pathological characteristics of study participants. The mean (SD) BMI of the study participants was 25.7 (± 4.8) Kg/M$^2$, with the majority of them, 132 (47.1), being within the normal weight category. There were 4 (1.4%) and 52 (18.6%) of the participants who were underweight and obese, respectively. The majority of the participants, 186 (66.4%), were immunologically responding to treatment ≥ 500 cells/mm$^3$, while 246 (87.9%) were currently virally suppressed < 50 copies/mL. There were 26 (9.3%) and 52 (18.6%) with elevated liver enzymes (ALT and AST), respectively. About 83 (29.4%) participants had HB < 13 (g/dL), considered anemic, while 141 (50.4%) had creatinine levels >0.8mg/dL, considered elevated (Table 2)

**Table 2. Summary of patient Immuno-pathological laboratory outcomes.**

| Variables | All Patients | COMORBIDITY TYPE | | | | | | | |
|---|---|---|---|---|---|---|---|---|---|
| | | HIV/TB | HIV/MDR-TB | HIV/T2D | HIV/HPT | HIV/T2D/HPT | HIV/TB/T2D | HIV/TB/HPT | HIV/TB/T2D/HPT |
| | n (280) | n (%) 58(20.7) | n (%) 6(2.1) | n (%) 42(15) | n (%) 33(11.8) | n (%) 18(6.4) | n (%) 9(3.2) | n (%) 9(3.2) | n (%) 4 (1.4) |
| **Body mass index (Kg/m²)** | | | | | | | | | |
| *Mean (± SD)* | 25.7 (± 4.8) | | | | | | | | |
| <18.5 (Underweight) | 4 | 0 | 0 | 1 (25) | 0 | 0 | 0 | 0 | 0 |
| 18.5 to 24.9 (Normal weight) | 132 | 25 (18.9) | 4 (3) | 21 (16) | 66 (50) | 10 (8) | 5 (3.8) | 3 (2.3) | 2 (1.5) |
| 25 to 29.9 (Overweight) | 92 | 16 (17.4) | 1 (1.1) | 16 (17) | 17 (18) | 1 (1) | 2 (2.2) | 2 (2.2) | 1 (1.1) |
| ≥30 (Obese) | 52 | 17 (32.7) | 1 (1.9) | 4 (8) | 10 (19) | 7 (13) | 2 (3.8) | 4 (7.7) | 1 (1.9) |
| *P value* | | 0.091 | 0.781 | 0.393 | 0.858 | 0.470 | 0.883 | 0.243 | 0.972 |
| **Baseline CD4+ (cell/µL)** | | | | | | | | | |
| Median (IQR) | 424.5 (301.5 - 531.5) | | | | | | | | |
| 1-500 | 186 | 38 (20.4) | 6 (3.2) | 29 (16) | 24 (13) | 14 (8) | 5 (2.7) | 4 (2.2) | 2 (1.1) |
| >501 | 94 | 20 (21.3) | 0 | 13 (14) | 9 (10) | 4 (4) | 4 (4.3) | 5 (5.3) | 2 (2.1) |
| *P value* | | 0.869 | 0.078 | 0.697 | 0.415 | 0.292 | 0.483 | 0.156 | 0.483 |
| **HIV viral load (Cells/mls)** | | | | | | | | | |
| *Mean (± SD)* | 6404.8 (± 29077.7) | | | | | | | | |
| 1-1000 | 246 | 44 (17.9) | 0 | 41 (17) | 32 (13) | 18 (7) | 9 (3.7) | 8 (3.3) | 4 (1.6) |
| >1001 | 34 | 14 (41.2) | 6 (17.6) | 1 (3) | 1 (3) | 0 | 0 | 1 (2.9) | 0 |
| *P value* | | **0.002** | **0.001** | **0.036** | 0.088 | 0.103 | 0.257 | 0.923 | 0.454 |
| **Baseline ALT (U/L)** | | | | | | | | | |
| Median (IQR) | 24 (19 – 34 ) | | | | | | | | |
| <56 | 254 | 54 (21.3) | 6 (2.4) | 38 (15) | 28 (11) | 16 (6) | 9 (3.5) | 8 (3.1) | 4 (1.6) |
| ≥56 | 26 | 4 (15.4) | 0 | 4 (15) | 5 (19) | 2 (8) | 0 | 1 (3.8) | 0 |
| *P value* | | 0.481 | 0.428 | 0.954 | 0.216 | 0.783 | 0.329 | 0.923 | 0.454 |
| **Baseline AST (U/L)** | | | | | | | | | |
| Median (IQR) | 25.3 (20.5 - 38) | | | | | | | | |
| <40 | 228 | 49 (21.5) | 3 (1.3) | 34 (15) | 25 (11) | 12 (5) | 6 (2.6) | 6 (2.6) | 1 (0.4) |
| ≥40 | 52 | 9 (17.3) | 3 (5.8) | 8 (15) | 8 (15) | 6 (12) | 3 (5.8) | 3 (5.8) | 3 (5.8) |
| *P value* | | 0.502 | **0.045** | 0.931 | 0.372 | 0.095 | 0.247 | 0.247 | **0.003** |
| **Baseline HB (g/dL)** | | | | | | | | | |
| Median (IQR) | 14.3 (12.8 - 15.5) | | | | | | | | |
| <13 | 83 | 16 (19.3) | 1 (1.2) | 9 (11) | 8 (10) | 4 (5) | 3 (3.6) | 3 (3.6) | 0 |
| ≥13 | 197 | 42 (21.3) | 5 (2.5) | 33 (17) | 25 (13) | 14 (7) | 6 (3) | 6 (3) | 4 (2) |
| *P value* | | 0.7 | 0.482 | 0.206 | 0.47 | 0.476 | 0.805 | 0.805 | 0.191 |
| **Creatinine** | | | | | | | | | |
| Median (IQR) | 0.9 (0.6 - 1.1) | | | | | | | | |
| <0.8mg/dL | 139 | 26 (18.7) | 1 (0.7) | 24 (17) | 17 (12) | 8 (6) | 4 (2.9) | 4 (2.9) | 1 (0.7) |
| >0.8mg/dL | 141 | 32 (22.7) | 5 (3.5) | 18 (13) | 16 (11) | 10 (7) | 5 (3.5) | 5 (3.5) | 3 (2.1) |
| *P value* | | 0.41 | 0.095 | 0.292 | 0.819 | 0.604 | 0.751 | 0.751 | 0.321 |

T2D- Type 2 Diabetes, TB – Tuberculosis, HTP- Hypertension, MDR – Multidrug-resistant, n - number; % - percentage; IQR - Interquartile range; SD Standard Deviation.

## Prevalence of TB and NCDs among PLWH

Of the participants, 179 (n = 280; 63.9%) were HIV mono-infected. The following comorbidities were identified: 58 (n = 280; 20.7%) HIV/TB coinfection, 42 (n = 280; 15%) HIV/T2D, 33 (n = 280; 11.8%) HIV/HPT, 18 (n = 280; 6.4%) HIV/T2D/HPT, 9 (n = 280; 3.2%) HIV/TB/T2D, 9 (n = 280; 3.2%) HIV/TB/HPT, and 4 (n = 280; 1.4%) HIV/TB/T2D/HPT. There were 6 (n = 280; 2.1%) PLWH coinfected with multidrug-resistant TB (MDR-TB). However, there were no PLWH with multidrug-resistant TB coinfected with either diabetes or hypertension (Fig 1).

## Comparison between baseline characteristics and HIV comorbidities

Univariate analysis of baseline characteristics revealed that participants' age, previous hospital admissions, missing ART or non-adherence, prior diagnoses of tuberculosis, diabetes, and hypertension, family histories of diabetes and hypertension, and viral load were associated with TB, HPT, and T2D comorbidities among PLWH (Table 1 and 2).

## Factors associated with the comorbidity of TB and Lifestyle diseases among PLWH

Table 3 summarizes the baseline factors associated with NCDs and TB comorbidity among study participants. In multivariate analysis of all the various possible comorbidity combinations including HIV/TB, HIV/T2D, HIV/HPT, HIV/MDR TB, HIV/TB/T2D, HIV/TB/HPT, HIV/T2D/HPT, and HIV/TB/T2D/HPT and baseline variables, only HIV/TB comorbidity was found significant. In a multivariate model, participants who were currently taking ARV only were less likely to have HIV/TB coinfection than those taking ARV plus other medications (aOR 0.5; 95% CI 1.4 – 0.6, p = 0.0001). Further participants who were virological suppressed were less likely to have HIV/TB coinfection than those who had virological failure (uOR 0.8; 95% CI 0.6 – 0.9, p = 0.017). Participants who were previously admitted to the hospital for an ailment remained more likely to have HIV/TB coinfection than those without previous hospital admission (aOR 1.2; 95%CI 1.1 – 1.4, p = 0.049). Participants who had previous TB infection remained more likely to have HIV/TB coinfection than those without TB diagnosis (uOR 1.6; 95%CI 1.0 – 3.0, p = 0.034).

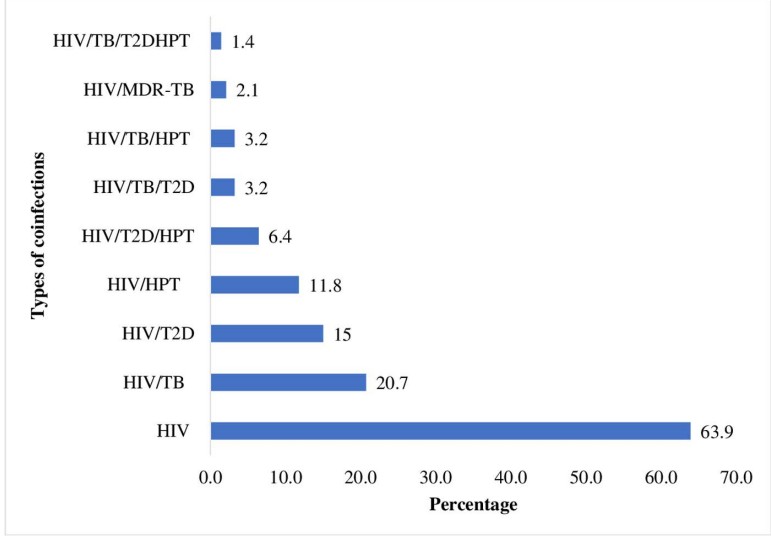

**Fig 1. The distributions in the types of comorbidities among study participants.**

**PLOS Global Public Health**

**Table 3. Bivariable and multivariable logistic regression analysis for the covariates of HIV/TB comorbidity (n = 280 patients).**

| Variables | Total | n | % | HIV/ TB | | | |
|---|---|---|---|---|---|---|---|
| | | | | Bivariate | P - value | Multivariate | P - value |
| | | | | uOR (95% CI) | | aOR (95% CI) | |
| **Age Group** | | | | | | | |
| 18 -30 | 82 | 12 | 15 | 0.8(0.2 - 2.4) | 0.651 | 0.8(0.2 - 2.5) | 0.691 |
| 31 - 40 | 121 | 27 | 22 | 0.8(0.3 - 2.6) | 0.726 | 0.8(0.3 - 2.6) | 0.778 |
| 41 - 50 | 53 | 11 | 21 | 0.8(0.3 - 2.7) | 0.714 | 0.8(0.3 - 2.7) | 0.743 |
| ≥51 | 24 | 7 | 29 | Referent | Referent | Referent | Referent |
| **Gender** | | | | | | | |
| Male | 146 | 29 | 20 | 0.9(0.8 - 1.2) | 0.892 | 0.9(0.8 - 1.2) | 0.879 |
| Female | 134 | 29 | 22 | Referent | Referent | Referent | Referent |
| **Education level** | | | | | | | |
| Primary | 47 | 8 | 17 | 1.1(0.7 - 1.4) | 0.978 | 1.0(0.7 - 1.4) | 0.966 |
| Secondary | 88 | 23 | 26 | 1.1(0.8 - 1.4) | 0.558 | 1.1(0.8 - 1.4) | 0.587 |
| Tertiary | 54 | 12 | 22 | 1.0(0.8 - 1.4) | 0.759 | 1.0(0.8 - 1.4) | 0.813 |
| Non-Formal | 91 | 15 | 16 | Referent | Referent | Referent | Referent |
| **Duration living with HIV (Years)** | | | | | | | |
| 1–5 | 94 | 22 | 23.4 | 0.9(0.6 - 1.5) | 0.848 | 1.2(0.7 - 1.5) | 0.898 |
| 6–10 | 165 | 30 | 18.2 | 0.9(0.6 - 1.4) | 0.682 | 1.2(0.7 - 1.5) | 0.782 |
| > 11 | 21 | 6 | 28.6 | Referent | Referent | Referent | Referent |
| **ARV regimen** | | | | | | | |
| ARV only | 222 | 2 | 0.9 | **0.5(0.4 - 0.6)** | **0.0001** | **0.5(0.4 - 0.6)** | **0.0001** |
| ARV plus other medications | 58 | 56 | 97 | Referent | Referent | Referent | Referent |
| **Changed ARV** | | | | | | | |
| Yes | 196 | 46 | 23.5 | 1.0(0.8 - 1.3) | 0.87 | 1.3(0.6 - 1.7) | 0.988 |
| No | 84 | 12 | 14.3 | Referent | Referent | Referent | Referent |
| **Hospital Admission** | | | | | | | |
| Yes | 28 | 12 | 42.9 | **1.2(1.1 - 1.7)** | **0.004** | **1.2(1.1 - 1.4)** | **0.049** |
| No | 252 | 46 | 18.3 | Referent | Referent | Referent | Referent |
| **Missed taking current ART** | | | | | | | |
| Yes | 104 | 23 | 22.1 | 1.1(0.8 - 1.3) | 0.87 | 1-1(0.8 - 1.3) | 0.857 |
| No | 176 | 35 | 19.9 | Referent | Referent | Referent | Referent |
| **Duration of cough experienced** | | | | | | | |
| <3 Weeks | 143 | 33 | 23.1 | 1.1(0.6 - 1.9) | 0.631 | 1.1(0.7 - 1.9) | 0.629 |
| 4-10 Weeks | 124 | 24 | 19.4 | 1.1(0.7 - 1.9) | 0.713 | 1.1(0.6 - 1.9) | 0.714 |
| >11 Weeks | 13 | 1 | 7.7 | Referent | Referent | Referent | Referent |
| **Previous tuberculosis diagnosis** | | | | | | | |
| Yes | 6 | 6 | 100 | **1.6(1.0 - 2.9)** | **0.049** | **1.6(1.1 - 3.0)** | **0.034** |
| No | 274 | 52 | 19 | Referent | Referent | Referent | Referent |
| **Previous diabetes diagnosis** | | | | | | | |
| Yes | 22 | 4 | 18.2 | 0.9(0.7 - 1.5) | 0.91 | 0.9(0.6 - 1.5) | 0.876 |
| No | 258 | 52 | 20.2 | Referent | Referent | Referent | Referent |
| **Body mass index (Kg/m²)** | | | | | | | |
| <18.5 (Underweight) | 4 | 0 | 0 | ND | ND | ND | ND |
| 18.5 to 24.9 (Normal weight) | 132 | 25 | 18.9 | 0.7(0.3 - 1.0) | 0.461 | 0.9(0.7 - 1.2) | 0.507 |
| 25 to 29.9 (Overweight) | 92 | 16 | 17.4 | 0.9(0.7 - 1.2) | 0.438 | 0.9(0.7 - 1.2) | 0.526 |
| ≥30 (Obese) | 52 | 17 | 32.7 | Referent | Referent | Referent | Referent |

*(Continued)*

**Table 3.** (Continued)

| Variables | Total | n | % | HIV/ TB Bivariate uOR (95% CI) | P - value | Multivariate aOR (95% CI) | P - value |
|---|---|---|---|---|---|---|---|
| **Baseline CD4+ (cell/µL)** | | | | | | | |
| 1-500 | 186 | 38 | 20.4 | 0.9(0.8 - 1.2) | 0.951 | 1.0(0.8 - 1.2) | 0.899 |
| >501 | 94 | 20 | 21.3 | Referent | Referent | Referent | Referent |
| **HIV viral load (Cells/mls)** | | | | | | | |
| 1-1000 | 246 | 44 | 17.9 | **0.04(0.05 - 0.06)** | **0.048** | **0.8(0.6 - 0.9)** | **0.017** |
| >1001 | 34 | 14 | 41.2 | Referent | Referent | Referent | Referent |
| **Baseline ALT (U/L)** | | | | | | | |
| <56 | 254 | 54 | 21.3 | 1.1(0.7 - 1.5) | 0.795 | 1.1(0.7 - 1.5) | 0.799 |
| ≥56 | 26 | 4 | 15.4 | Referent | Referent | Referent | Referent |
| **Baseline AST (U/L)** | | | | | | | |
| <40 | 228 | 49 | 21.5 | 1.1(0.8 - 1.4) | 0.804 | 1.0(0.8 - 1.4) | 0.797 |
| ≥40 | 52 | 9 | 17.3 | Referent | Referent | Referent | Referent |
| **Baseline HB (g/dL)** | | | | | | | |
| <13 | 83 | 16 | 19.3 | 0.9(0.8 - 1.2) | 0.887 | 0.9(0.8 - 1.2) | 0.915 |
| ≥13 | 197 | 42 | 21.3 | Referent | Referent | Referent | Referent |
| **Creatinine** | | | | | | | |
| <0.8mg/dL | 139 | 26 | 18.7 | 0.9(0.8 - 1.2) | 0.761 | 0.9(0.8 - 1.2) | 0.764 |
| >0.8mg/dL | 141 | 32 | 22.7 | Referent | Referent | Referent | Referent |

n - number; % - percentage; uOR – Unadjusted odd ratio; aOR – adjusted odd ratio.

## Discussion

In the general population globally, NCDs are among the top ten public health problems. They significantly impact PLWH, so curbing their rising burden in developing countries is urgent and could reduce the morbidity and mortality of these diseases. This can be achieved by regularly monitoring the burden of NCDs and translating these findings to improve or institute control programs. This study provides additional information on NCDs and TB among PLWH in eastern Kenya, where the statistic is skewed.

The comorbidity of NCDs and TB among PLWH varied widely. A prevalence of 20.7% was reported for HIV and TB comorbidity, which is lower than the 37.9% reported in Ethiopia [17], and 30.1% in a study in South Africa [18]. The current study reported a higher prevalence of TB/HIV comorbidity than 3.6% reported in India [19], 11% in Tanzania [20], 2.5% in Mexico, and 5.6% in the Netherlands [21,22]. There were 2.1% PLWH coinfected with MDR-TB. This study did not report MDR-TB coinfection with either T2D or HPT. Multi-drug-resistant TB continues to be a global public health challenge, particularly in sub-Saharan Africa, increasing the weight of other communicable and non-communicable diseases ravaging the region [23,24]. In 2021, WHO estimated the global burden of new TB cases with MDR/RR-TB at 3.6%, with 18% of them treated [24]. In a meta-analysis of Salari et al., [25], the global pooled prevalence of multidrug-resistant, Isonazid, Rifampicin, and extensively drug-resistant TB were calculated as 11.6%, 15.7%, 9.4%, and 2.5%, respectively. In 2021, WHO further listed India, the Russian Federation, and Pakistan as accounting for 26%, 8.5%, and 7.9%, respectively, of global cases of drug-resistant TB burden [24]. In Pakistan Ali et al., [26] reported a prevalence of 4.9% MDR-TB cases, while in Nigeria, a prevalence of 3.5% MDR-TB was reported [27,28]. In Saudi Arabia, Sambas et al., [2020] reported a prevalence of 5% MDR TB among PLWH. The prevalence of MDR-TB seems heterogeneous, but the burden seems to be more in countries categorized by WHO as such [24].

MDR-TB reported in this region of Kenya reiterates the need to strengthen further monitoring of TB and MDR-TB within the national HIV programs.

About 15% of the study patients had HIV/T2D comorbidity, which is comparable to 13.7% in Vietnam and 13.5% in Ethiopia [29,30]. Much lower HIV/T2D coinfection rates of 11.5%, 7%, 6.1%, and 5.5% have been reported in Ethiopia, Sweden, South Africa, and Mali, respectively [31–34]. This study reported a prevalence of 11.8% for HIV/HPT coinfection, which was lower than the 25.5% reported in South Africa [33], 24.8% in Uganda [35], 31.7% in Nigeria [36], 36% in France [37], and 38% in the USA [38]. Generally, a low prevalence of hypertension and HIV coinfection has also been reported, such as 8% in Cote d'Ivoire [39]. These disproportions in the prevalence of HPT among PLWH are a pointer to the heterogeneous nature of traditional risks associated with HPT in these nations.

The current prevalence of 6.4% HIV/T2D/HPT coinfection was lower than 22.6%, 19.6%, and 19% reported in South Africa, Zimbabwe, and Ethiopia, respectively [15,13]. The prevalence of HPT and T2D among PLWH is on the rise among the African population and is the leading risk factor for mortality [40,13]. There were 3.2% PLWH coinfected with TB/T2D, which was lower than the 5.8% and 4.8% reported in Cameron and Ethiopia, respectively [41,31]. From the literature, this is among the few studies highlighting the burden of HIV/TB/T2D coinfection in Africa. Patients with T2D were 3.6 times more likely to develop active TB infection [42]. The high rates of T2D and HIV coinfection among patients with TB are partly because HIV jeopardizes the immune makeup and progresses from latent *M. tuberculosis* to active TB [43].

There were 3.2% PLWH coinfected with TB/HPT. Previously, in Western Kenya, a higher prevalence of HIV/TB/HPT of 11.2% and 7.4% were reported in men and women, respectively [44], with 12.5% reported in Tanzania [45]. A lower rate of 1.6% was reported in South Africa [46]. This study further showed that 1.4% of PLWH had a quadruple comorbidity of TB/T2D and HPT. Reports are sparse examining the quadruple coinfection of NCDs and TB among the general population and PLWH. In the UK, Lorenc et al., [47] reported a mean number of 1.1 and 1.4 of general and HIV-associated comorbidities amongst HIV patients, respectively. The study identified diabetes, hypertension, and TB comorbidities among patients with HIV. These studies show significant heterogeneity in the prevalence of TB and NCDs among PLHWs. The difference in these rates could partly be explained by the overall prevalence of TB and HIV, which is higher in developing countries, including Kenya. HIV-infected persons are 15–22 times more likely to develop active TB than HIV-negative persons [48]. In any one patient, *M. tuberculosis* and HIV energize one another, hastening the degradation of immune functions [1,49]. Older age, being male, lengthened duration living with HIV, low CD4 levels, viral failure, high body mass index, lower socio-economic status, ethnicity, and cultural background are among the factors associated with the development of NCDs among PLWH [50].

In multivariate models, significant associations were found only between evaluated correlates and HIV/TB comorbidity. The PLWH receiving ARV medication only was protected from TB coinfection compared to those taking ARVs in combination with other drugs. This is similar to previous studies showing timely access to ART minimizes immune deterioration and improves TB outcomes [51]. Initiating HAART during TB treatment is problematic due to overlaying toxicities, drug-drug interactions, and a high number of pills taken, which may reduce adherence [52]. Delaying HAART may lead to prolonged or declining immune suppression. Balancing these risks when deciding when to initiate HAART with early treatment initiation reduces morbidity and mortality.

Virological suppression was protective against TB/HIV coinfection, similar to results in Turkey [53] and South Africa [54]. The association of virologic failure on ART and a higher risk for TB development has been shown [55]. It may be because HIV accelerates the loss of CD4 + T lymphocytes and promotes the progression [53].

Previous hospital admissions were linked to TB coinfection among PLWH. In Congo, Shah et al. [19] demonstrated that TB/HIV coinfection was associated with adverse medical outcomes, such as virologic failure that required hospitalization and loss of patient follow-up leading to death. In India, hospital admissions contributed to TB coinfection among PLWH; conversely, HIV/TB coinfection necessitated hospitalization. The study indicated that most patients showed improved clinical outcomes after discharge, although approximately 3% died during their hospital stay [56,57]. The report highlighted a

significant 15% mortality rate among those with TB/HIV coinfection during hospitalization [56,57]. Hospitalization typically results from deteriorating health conditions. Patients admitted due to TB/HIV coinfection face an elevated risk of death, which may stem from treatment failure, non-adherence to anti-TB medications, or a late diagnosis of TB and/or HIV [58].

Previous TB infection was linked to HIV/TB coinfection. Horsburgh et al. [59] demonstrated that a past TB infection could, on the one hand, provide substantial protection against TB reinfection while simultaneously noting that 36–79% of TB disease cases can be traced back to the already-infected population. Higher rates of reactivation and reinfection are prevalent in countries with a high incidence of TB [60].

Some additional studies have identified factors associated with TB/HIV co-infection that the current study did not observe, including underlying diseases, immunosuppressive agents, substance abuse, smoking, and various behavioral, social, and environmental factors [53]. The ratio of neutrophils to lymphocytes has also been shown to predict TB among people living with HIV/AIDS (PLWHA) [61].

## Conclusions

This study indicates that eastern Kenya is facing a syndemic of NCDs and TB, including drug-resistant strains among PLWH. Consistent adherence to ART is crucial for achieving viral suppression, and these treatments also provide protection against NCDs and TB in PLWH. The findings highlight the necessity of integrating NCD management with HIV and TB treatment programs in Kenya.

## Supporting information

**S1 Data. Study related data.**
(XLSX)

## Acknowledgments

The authors wish to acknowledge all the study patients and staff of Nyambene Sub-County Hospital in Eastern Kenya and the Center for Microbiology Research, Kenya Medical Research Institute.

## Author contributions

**Conceptualization:** Patrick Kiogora Muriuki, Moses Njire, Perpetual Ndung'u.

**Data curation:** Patrick Kiogora Muriuki, Musa Otieno Ngayo, Juster Mungiria, Winfred Asiko Nyanya, Daniel Owuor.

**Formal analysis:** Musa Otieno Ngayo, Juster Mungiria, Daniel Owuor.

**Funding acquisition:** Patrick Kiogora Muriuki.

**Investigation:** Patrick Kiogora Muriuki, Moses Njire, Winfred Asiko Nyanya, Perpetual Ndung'u.

**Methodology:** Patrick Kiogora Muriuki, Juster Mungiria, Winfred Asiko Nyanya, Daniel Owuor.

**Project administration:** Perpetual Ndung'u.

**Resources:** Moses Njire.

**Supervision:** Musa Otieno Ngayo, Moses Njire, Perpetual Ndung'u.

**Validation:** Patrick Kiogora Muriuki, Musa Otieno Ngayo, Juster Mungiria, Winfred Asiko Nyanya, Daniel Owuor, Perpetual Ndung'u.

**Writing – original draft:** Patrick Kiogora Muriuki, Musa Otieno Ngayo.

**Writing – review & editing:** Patrick Kiogora Muriuki, Moses Njire, Juster Mungiria, Winfred Asiko Nyanya, Daniel Owuor, Perpetual Ndung'u.

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
