## [Decision Letter · Decision Letter 0]

PGPH-D-25-00036

Non-communicable diseases and resistant tuberculosis, a growing burden among people living with HIV in Eastern Kenya

Dear Dr. Ngayo,

Thank you for submitting your manuscript to PLOS Global Public Health. After careful consideration, we feel that it has merit but does not fully meet PLOS Global Public Health’s publication criteria as it currently stands. Therefore, we invite you to submit a revised version of the manuscript that addresses the points raised during the review process.

We look forward to receiving your revised manuscript.

Kind regards,

Lei Gao

Academic Editor

Journal Requirements:

Additional Editor Comments (if provided):

Reviewers' comments:

**Comments to the Author**

1. Does this manuscript meet PLOS Global Public Health’s publication criteria?

Reviewer #1: Yes

Reviewer #2: Yes

2. Has the statistical analysis been performed appropriately and rigorously?

Reviewer #1: Yes

Reviewer #2: Yes

3. Have the authors made all data underlying the findings in their manuscript fully available (please refer to the Data Availability Statement at the start of the manuscript PDF file)?

Reviewer #1: Yes

Reviewer #2: Yes

4. Is the manuscript presented in an intelligible fashion and written in standard English?

Reviewer #1: Yes

Reviewer #2: Yes

Reviewer #1: This is important work and I appreciate the focus on the overlap.

I would focus more in the introduction and conclusion and some analysis on 1) diabetes being a risk for TB (one of the leading risks, replacing malnutrition and HIV in some locations) and a risk for more severe outcomes

https://www.who.int/news/item/07-02-2025-who-operational-handbook-on-tb-and-comorbidities--diabetes#:~:text=According%20to%20the%202024%20WHO,a%20high%20prevalence%20of%20TB.

https://pmc.ncbi.nlm.nih.gov/articles/PMC11342417/

https://pmc.ncbi.nlm.nih.gov/articles/PMC10662654/

and HIV as a risk for HTN, which is more complicated and may be mediated by weight gain with ARVs, but also persistent HIV is a risk for Hypertension and it can additionally be mediated by the virus' impact on the cardiovascular and renal system directly, which leads to HTN risks.

https://pmc.ncbi.nlm.nih.gov/articles/PMC7877548/

https://academic.oup.com/ofid/article/12/Supplement_1/ofae631.773/7988171?login=false

https://pmc.ncbi.nlm.nih.gov/articles/PMC11886560/

https://pmc.ncbi.nlm.nih.gov/articles/PMC6002926/

Additionally TB, especially if advanced, can lead to HTN, if

I would use this focus to inform further statistical analyses. There are different approaches which can be used to improve the statistical analysis including methods of considering the likely causal relationships here.

You could consider expanding the multivariable analysis to include all comorbidity combinations instead of just HIV/TB, which would provide more comprehensive insights into these complex disease interactions. The analysis would benefit from including interaction terms (e.g., Diabetes × TB severity, HIV × Hypertension) to examine whether these conditions have multiplicative rather than just additive effects. Stratifying analyses by HIV status and ART duration would better assess how diabetes affects TB risk differently in various patient subgroups. Time-to-event models, whenever possible, would provide a more dynamic analysis of how diabetes duration affects TB risk and how ART duration influences hypertension incidence. Applying propensity score matching would help balance baseline characteristics between comparison groups, minimizing selection bias and strengthening causal inferences from this observational study.

Would reword this "6.4% HIV/T2D/HPT coinfection" and "3.2% PLWH coinfected with TB/HPT" and "quadruple coinfection of NCDs and TB" and "Triple coinfections included 18 (6.4%) HIV/T2D/HPT,

37 9 (3.2%) HIV/TB/T2D, and 9 (3.2%) HIV/TB/HPT, with a quadruple coinfection of

38 HIV/TB/T2D/HPT among 4 (1.4%) participants." etc Hypertension and diabetes are not infections.

Would capitalize differently: coronavirus (coViD-19) pandemic,

Reviewer #2: Overall, the statistical analysis is done well, findings are well presented, and the manuscript is well written and reads well. Only minor corrections required in the background and methodology. Thank you!

**Do you want your identity to be public for this peer review?** For information about this choice, including consent withdrawal, please see our Privacy Policy

Reviewer #1: No

Reviewer #2: **Yes: ** Sthabiso Bohlela

---

## [Editor Report · Decision Letter 1]

Non-communicable diseases and resistant tuberculosis, a growing burden among people living with HIV in Eastern Kenya

PGPH-D-25-00036R1

Dear Dr. Ngayo,

We are pleased to inform you that your manuscript 'Non-communicable diseases and resistant tuberculosis, a growing burden among people living with HIV in Eastern Kenya' has been provisionally accepted for publication in PLOS Global Public Health.

Best regards,

Lei Gao

Academic Editor